# Intake of Processed Meat and Association with Sociodemographic and Lifestyle Factors in a Representative Sample of the Swiss Population

**DOI:** 10.3390/nu11112556

**Published:** 2019-10-23

**Authors:** Janice Sych, Ivo Kaelin, Fabienne Gerlach, Anna Wróbel, Thu Le, Rex FitzGerald, Giulia Pestoni, David Faeh, Jean-Philippe Krieger, Sabine Rohrmann

**Affiliations:** 1Institute of Food and Beverage Innovation, ZHAW School of Life Sciences and Facility Management, Einsiedlerstrasse 34, 8820 Wädenswil, Switzerland; fabi.kretz@gmail.com (F.G.); tlthi@student.ethz.ch (T.L.); 2Institute of Applied Simulation, ZHAW School of Life Sciences and Facility Management, Einsiedlerstrasse 31a, 8820 Wädenswil, Switzerland; ivo.kaelin@zhaw.ch (I.K.); wrob@zhaw.ch (A.W.); 3Swiss Centre for Applied Human Toxicology (SCAHT) University of Basel, Missionstrasse 64, 4055 Basel, Switzerland; rex.fitzgerald@unibas.ch; 4Division of Chronic Disease Epidemiology, Epidemiology, Biostatistics and Prevention Institute, University of Zurich, Hirschengraben 84, 8001 Zurich, Switzerland; Giulia.Pestoni@usz.ch (G.P.); david.faeh@bfh.ch (D.F.); jean-philippe.krieger@gu.se (J.-P.K.); sabine.rohrmann@uzh.ch (S.R.); 5Health Department–Nutrition and Dietetics, Bern University of Applied Sciences, 3008 Bern, Switzerland

**Keywords:** processed meat, meat products, meat intake, meat consumption, chronic disease, menuCH, nutrition survey

## Abstract

Processed meat (PM) intake is associated with health risks, but data are lacking in Switzerland. Using national representative data from a recent menuCH Survey, we first aimed to quantify intake of PM and its subtypes, and second to investigate associations with sociodemographic and lifestyle factors by multivariable regression analysis. PM was consumed by 72% of the population, and mean daily intake was 42.7 g/day (standard error of the mean (SEM) 1.2 g/day), ranging considerably across PM subtypes: highest intake of sausages 18.1 g/day (SEM 0.7 g/day) and lowest of bacon 2.0 g/day (SEM 0.2 g/day). PM intake by women was 4.7 g/1000 kcal lower than men (95% confidence interval (CI): −6.7; −2.7) and 2.9 g/1000 kcal lower in the French- language region compared with the German region (95% CI: 2.4; 8.7). Among sociodemographic and lifestyle factors examined, BMI (obese vs. normal: 5.5 g/1000 kcal, 95% CI: 2.4; 8.7) and current smoking (vs. never smoked: 3.1 g/kcal, 95% CI: 0.6; 5.6) were independently associated with PM intake. The results are a first description of PM intake, separate from other meat types, and which identified associations with two unhealthy lifestyle factors in Switzerland. Such data will contribute to better nutritional recommendations and guidance for public health interventions.

## 1. Introduction

In Switzerland, processed meat (PM) has a unique role in diet, closely linked to culture and tradition, and is often the main course at festive gatherings. However, concerns for this food group are growing due to epidemiological evidence for its association with chronic diseases such as coronary heart disease, several types of cancers and diabetes type 2 [1,2,3], and due to emerging priorities for sustainable nutrition [4,5]. On a global scale, processed meat intakes have been quite stable (1990–2010), but with large variations between countries, from 2.5 to 66.1 g/day [4].

PM products are whole pieces or mixtures of comminuted meats consisting of mainly pork and beef, less often poultry, and with other animal parts [6]. Processing steps vary widely and may include salting, curing, cooking, fermentation, or smoking to improve color, flavor, and shelf-life. In Switzerland, technical specifications for PM products are compiled in an industry manual and serve to standardize composition, processing, and quality [7]. Despite some differences in definitions, typical PM categories used across Europe are ham, bacon, sausages, and other PM types (e.g., ready-to-eat meat products) [8].

Processed meat is associated with higher health risks than fresh, unprocessed meats (UPM) [1,2,9]. Typically, PM has lower levels of protein and iron, especially compared with red UPM, and has higher total and saturated fats [10]. Another important difference is that both red and white PM contain salt at high levels, as well as preservatives such as nitrate and nitrite which are introduced during processing [11]. Consumption of PM and red meat is associated with higher risk of colorectal cancer (CRC), which led to the recent classification of PM as carcinogenic to humans (Group 1) and red meat as probably carcinogenic (Group 2A) by the International Agency for Research on Cancer [3]. Worldwide, CRC is the second and third most common cancer in women and men, respectively. From 2011 to 2015, approximately 2400 men and 1900 women in Switzerland were diagnosed with CRC, leading to average death numbers of 950 and 750, respectively [12]. PM is also a dietary risk factor for cardiovascular disease which has a much higher burden with 9,725 and 11, 972 deaths per year (2016), men and women, respectively [13]. From a public health perspective, the assessment of PM intakes at population level is therefore an important priority worldwide. 

It is also crucial to gain insights about factors which influence PM consumption, and thus have possible impacts on health and on sustainable nutrition. Many national surveys and studies have shown that men have higher consumption of all types of meat than women [4,14,15,16]. Additionally, UPM and PM consumption were each associated with higher BMI, smoking, lower education levels, and increased age in the European Prospective Investigation into Cancer and Nutrition [17]. Meat intake was inversely associated with a sociodemographic index consisting of education, income, and occupation, but the association was not the same for different meat types. For example, lower economic status was more strongly associated with intake of PM than with UPM [4].

Nutritional data on PM intake of the Swiss population are few and based on methods which did not separately quantify it from UPM [18]. The National Nutrition Survey menuCH provides the first representative data on food and beverage consumption by adults from three main language regions, along with sociodemographic and lifestyle factors, and anthropometry measurements [19]. These data can therefore be used to gain insights into determinants of PM consumption in Switzerland. However, the menuCH survey did not include biological markers, and data cannot be linked to a cancer database. The first report from the survey revealed high intakes of total meat, regional differences in meat intake, and PM was the highest consumed food in the meat and fish category [19,20]. 

In view of the evidence-based health risks associated with PM intake, the current study examined the menuCH data more specifically for this meat type. We first aimed to describe the distribution of total meat intake as PM and UPM in Switzerland by sex, language region, and age-category. Then we focused more specifically on PM consumption by quantifying intakes of PM subtypes. Our second aim was to investigate the association between energy-standardized total PM intake and sociodemographic and lifestyle factors.

## 2. Materials and Methods 

### 2.1. Study Design 

The National Nutrition Survey menuCH was a cross-sectional population-based survey conducted from January 2014 to February 2015 in ten study centers of Switzerland. Recruitment of adults 18 to 75 years old was done randomly [19] using 35 strata from the seven major areas of Switzerland (Lake Geneva, Midlands, Northwest, Zurich, Eastern, Central, and Southern Switzerland) and from five age categories (18–29, 30–39, 40–49, 50–64, and 65–75 years old). Net participation rate was 38%, and 2086 participants comprised the final study group. A complete study description was reported earlier [19,21]. In the current study, we analyzed the data from 2057 participants who completed both 24 h dietary recalls (24HDR).

### 2.2. Ethical Approval

Ethical approval for the survey was obtained from the main ethics committee in Lausanne (Protocol 26/13, 12 February 2013) and by the corresponding regional ethics committees. Guidelines of the Declaration of Helsinki were respected, including written informed consent from the study participants. Registration information is: ISRCTN registration number 16778734 (https://doi.org/10.1186/ISRCTN16778734).

### 2.3. Dietary Assessment

A first 24 h dietary recall (24HDR) was conducted face-to-face, followed by a second telephone interview two to six weeks later [19]. Interviews were done by 15 trained dietitians in German, French or Italian, using the software GloboDiet^®^ (formerly EPIC-Soft^®^, version CH-2016.4.10, International Agency for Research on Cancer (IARC, Lyon, France) [22,23] adapted for Switzerland (GloboDiet^®^ trilingual databases dated 12 December 2016, IARC, Lyon, France and Federal Food Safety and Veterinary Office, Bern, Switzerland). During the interviews, a food-picture book was used to facilitate food reporting [24], and descriptive information about the consumption events (e.g., methods of preparation) was entered into the software.

A pilot survey was conducted, and results were used to assess and improve the standard operating procedures of the survey team [21]. Data cleaning and controls for inconsistencies in reported data followed IARC’s guidelines [23]. Other details of assessment were described earlier [19,21].

### 2.4. Sociodemographic, Lifestyle, and Anthropometric Variables 

Participants completed a questionnaire at the first 24HDR in French, German or Italian which queried on dietary habits and sociodemographic and lifestyle factors [22]. Many of the questions were taken from the Swiss Health Survey with modifications [18]. The following variables were investigated in the current analysis: nationality (Swiss only; Swiss with dual citizenship; other); education (primary; secondary; tertiary education); gross household income (<6000; between 6000 and 13,000; >13,000 Swiss Francs/month); household status at 6 levels (living alone; adult living with parents; one-parent family with children; couple without children; couple with children; others); smoking status (never; former; current); overall health status (regrouped from 5 into 2 levels: bad to medium; good to very good); and currently following a weight-loss diet (yes; no). All answers were self-declared. 

Four categories of age were used in the present analysis: 18–29; 30–44; 45–59; and 60–75 years old. Language region was determined by the canton of residence of participants (German: Aargau, Basel-Land, Basel-Stadt, Bern, Lucerne, St. Gallen, Zurich; French: Geneva, Jura, Neuchâtel, Vaud; Italian: Ticino). The physical activity of participants was assessed using the short-form International Physical Activity Questionnaire (IPAQ) and categorized as low, moderate, and high physical activity [25,26,27]. Measurements of body weight, height, and waist circumferences followed international standard protocols [28] and were used to calculate body mass index (BMI), except for pregnant (*n* = 14) and lactating (*n* = 13) women (self-reported values of weight before pregnancy) or when measurements were impossible (*n* = 7), as reported earlier [21]. 

### 2.5. Categorization of Processed Meats

Using the consumption data, the analysis was based on the subcategory of meat and meat products defined in Globodiet^®^ [19,22] with removal of meat substitutes and addition of meat from bolognaise sauce (from subcategory ‘Savory Sauces’). Intakes of 78 different PM products were assigned to four subcategories: ham, bacon, sausage, and other PM (Appendix A), similar to early studies in Europe [17]. The category of other PM included sausage meat skewers, ground meat, including that from bolognaise sauce (considered as 100% meat), and ready-to-eat meat products (chicken nuggets, sausage meat skewers, hamburgers, and luncheon meats). 

Technical specifications for PM products [7] were used to categorize PM intakes based on the presence or absence of nitrate and nitrite (nitrate/nitrite). When information was not given in the technical specifications, labels of similar retail products (at least 3) were checked for the E-numbers corresponding to nitrate/nitrite (in Switzerland E-249; E-251, respectively) by one scientific assistant [29]. Undefined meat consumptions were not included in this categorization, since it was not possible to check with similar retail products. The method was repeated, and results were reviewed by an expert in the Swiss meat industry.

### 2.6. Data Handling and Statistical Analysis

Using reported data from the two 24HDR, mean and standard error of the mean (SEM; g/day) were used to describe the consumption of the main categories of meat (total meat, PM, and UPM) for the population and by sex, language region, and age category. Figure 1 shows the proportions of PM and UPM consumed. Further analysis focused on PM consumption, including a description of the main subtypes, shown as average energy-standardized intakes (g/1000 kcal) to facilitate comparisons with other studies. The distribution of PM intake (crude data) for the population was examined by histograms and quartiles; and normality was checked by Shapiro-Wilk’s test for total PM and also for each PM subcategory [30]. 

Multivariable linear regression analysis was used to study associations between average energy-standardized PM intake (g/1000 kcal) and selected sociodemographic and lifestyle factors. We selected factors which are typically used in national surveys: sex, age and nationality; and specific to Switzerland, language region was used to investigate possible differences in PM intake between the regions. Economic indicators and other factors relevant for PM consumption were also included (education, household status, household income, and currently on a diet) and lifestyle factors with evidence-based health associations (smoking, BMI category, and physical activity level). Self-reported health was included, since it was shown to be a strong predictor of health [31]. Multivariate imputation by chained equations was used to provide estimates for missing values (*m* = 25).

All descriptive and statistical analyses were conducted with R (version 3.6.0). The mice package was used for the multivariate imputation [32]. Additional R-packages were used for calculating weighted SEM (SEM by radiant data) and figures (ggplot2, dplyr, and plyr).

### 2.7. Weighting of Data

Weighting factors were applied to all means and SEM and in the regression model to correct for sampling design and nonresponse. All results were weighted for age, sex, marital status, major area of Switzerland, nationality, and household size. Consumption data were also corrected for season and days of the week. Further details of the menuCH Survey weighting strategy are available in a public data repository [33].

### 2.8. Reporting Data

The STROBE-nut guidelines were used to report the findings of the present study [34].

## 3. Results

### 3.1. Characteristics of the Study Population

Table 1 describes the sociodemographic, lifestyle, and anthropometric characteristics of the study group, 2057 adults who completed two 24HDR. After weighting, the sample represented 4,627,878 individuals with balanced representation of men and women, and couples with and without children, and highest representation of middle ages (29.9%). The majority were Swiss citizens (61.4%), from the German-language region (69.2%) and with good to very good health status (87.1%). Considering lifestyle risk factors, 43.5% of the population had above-normal BMI, and 23.3% were current smokers.

PM consumption was reported by 72.0% of participants (3,331,232 participants, weighted data), who showed similar representation of the investigated sociodemographic and lifestyle variables compared with the total population. 

### 3.2. Intake of Total Meat, Processed and Unprocessed 

Figure 1 shows a mean of 108.9 g/day for total meat consumption of the population, distributed as 42.7 g/day of PM and 66.2 g/day of UPM. Men consumed considerably higher quantities of both meat categories compared to women. Standardizing PM intakes for energy intake confirmed a higher PM energy contribution for men compared to women, i.e., 21.9 and 15.4 g/1000 kcal, respectively (Table 2).

Intakes of UPM were highest in the French-language region, whereas PM intakes were highest in the German-language region (Figure 1). The youngest age category reported slightly higher intakes of both PM and UPM (Figure 1), confirmed by energy-standardized data (Table 2 for PM; UPM not shown). 

PM intake (and subcategories) were not normally distributed in the total population (Shapiro–Wilk’s test, *p* ≤ 0.05), shown in Appendix A. At both interviews, 589 study participants did not report PM intake, resulting in a considerably lower median than the mean (23.0 g/day and 41.9 g/day, respectively, crude data).

### 3.3. Intake of Different Types of Processed Meat

Among the four PM categories, sausages were consumed in highest quantities (mean intake 7.8 g/1000 kcal), followed by the category other PM, ham, and bacon (5.4, 4.6, and 0.8 g/1000 kcal, respectively) (Table 2). Ham consumption was highest in the Italian-language region, whereas sausage intake was highest in the German-language region. In all PM categories, men had higher intakes than women. 

Intakes of total PM, and subcategories of ham, bacon, and sausages were relatively stable across the age groups, with slightly higher total PM intake (20.4 g/1000 kcal) for the youngest age (18–29 years, Table 2). This age group also had slightly higher intake of other PM. Sausages were consumed in highest amounts, ranging from 7.3 to 8.4 g/1000 kcal across the age groups.

### 3.4. Processed Meats Containing Nitrate and Nitrite 

In Table 3, 87% of total PM intake was assigned to the category of PM with or without nitrate/nitrite. The remaining 13% of intake was excluded in this analysis, due to missing information about these preservatives. Intake of nitrate/nitrite-containing PM was three times higher than PM without (27.4 and 9.8 g/day, respectively) by population, and slightly higher in the German-language region (28.9 g/day).

### 3.5. Sociodemographic and Lifestyle Factors Associated with Processed Meat Intake

The variables sex, language region, education, following a weight-loss diet, BMI category, and smoking status were significant determinants of PM consumption (Table 4), but age group was not significant. Women consumed 4.7 g/1000 kcal less PM than men (95% confidence interval (CI): −6.7; −2.7), and participants following a weight-loss diet showed a reduced PM intake by 7.5 g/1000 kcal (CI: −11.7; −3.4) compared to those not following a diet. PM intake in the French-language region was 2.9 g/1000 kcal lower (95% CI: −5.2; −0.7) than in the German-language region. The same regression coefficient was obtained for participants with tertiary education compared with secondary education (Table 4). Overweight, obesity, and current smoking were positively associated with PM intake, 3.9 g/1000 kcal (95% CI: 1.6; 6.1), 5.5 g/1000 kcal (95% CI: 2.4; 8.7), and 3.1 g/kcal (95% CI: 0.6; 5.6), respectively.

## 4. Discussion

### 4.1. Summary of Main Findings

The current analysis of menuCH Survey data revealed that PM intake represents about 40% of total meat consumed, with a daily mean of 42.7 g/day for the Swiss population, 65% of which contained nitrate/nitrite. Sausage was the PM subtype consumed at highest levels by the population and especially in the German-language region. Energy-standardized PM intake was significantly lower for women, by participants in the French-language region, with higher education level, and following a diet. Our results revealed positive associations between total PM consumption and two unhealthy lifestyle variables, current smoking and high BMI.

### 4.2. Processed Meat Consumption in Switzerland and in Europe

Comparisons of our results with other studies in Switzerland is limited due to lack of meat consumption data for the population, and non-equivalent assessment methods. Conducted every five years, the Swiss Health Survey only collects frequency data on combined intakes of meat and meat products [16,18]. According to the 2012 survey, 41.5% of the population consumed meat or sausages one to three times per week [18]. Prior to the menuCH Survey, food balance sheets provided estimates of nutritional data in Switzerland, and total meat intake was estimated at 52 kg per person, approximately 142 g/day (2018) [35], considerably higher than the result of this analysis (Figure 1). Due to their characteristic meat mixtures and processing steps, this type of estimate for PM is not possible [36].

Mean intakes of PM were shown to vary considerably across Europe [37]. Calculated on 2000 kcal, the mean total PM intake in Switzerland was 37.3 g/day, which is lower than the equivalent energy-standardized mean intake in the Czech Republic (54.0 g/day) but higher than that in France (34.7 g/day), Denmark (27.3 g/day), and Italy (25.5 g/day) [37], all determined by dietary recalls from two to seven days. In France, mean intake of PM was 37.1 g/day determined by validated web-based 24HDR in the Nutrinet Study [38]. The mean total PM in the Irish population was 49 g/day and 38 g/day for age groups 18 to 64 years and over 65 years, respectively [39]. In the current study age group was not a significant determinant of energy-standardized PM intake in the Swiss population. 

### 4.3. Nutritional Recommendations for Meat and Processed Meat Consumption

The Swiss recommendation for meat intake refers to combined intakes of all meats, i.e., 2 to 3 weekly portions of total meat (100 to 120 g portion) and limitation of PM to once per week [40] but without mention of portion size. Therefore, comparisons of our data (Table 2) with these guidelines are quite limited. Assuming 15 g/day as daily reference, PM intake in Switzerland is 2.8 times too high [19]. Our data show that mean daily sausage intake (18.1 g/day) slightly exceeds this reference value. Compliance to the Swiss meat recommendation by men was reported to be low and decreased between 2009 and 2017 in French-speaking Switzerland [41]. These results were based on frequency data for combined intakes of meat and processed meat; therefore, no specific conclusions could be made about PM intake.

Other nutritional societies in Europe have set clearer limits for PM intake, such as 25 g/day or three weekly portions in France [42,43]; three weekly 50 g portions in Italy [44] or complete avoidance by the World Cancer Research Fund [45]. In France, PM appears at the top of the food pyramid, which emphasizes the need for low intake, and guidelines also include a warning linking too frequent PM consumption with increased risk of certain cancers. In contrast the Swiss Food Pyramid shows PM at the same level as other protein sources which have high biological value (i.e., meat, fish and eggs) [40]. The results of our study could impact nutritional policy in Switzerland [46] and especially provide a quantitative basis for revising the meat guideline. In line with current evidence, PM should be considered separately from other types of meat and in the food pyramid, presented as a discretionary food group with a daily limit of intake. A greater emphasis on nutritional composition might also be needed in these guidelines, especially for sausages where high levels of energy and saturated fats might contribute to obesity and other health risks [47,48,49]. 

### 4.4. Different Types of Processed Meat 

Quantifying intake of PM subtypes is important due to their differences in composition and processing, which could be linked to differences in health outcomes [8,11,50]. To date, very few studies have assessed the consumption of different PM subtypes [14,51] and even fewer on red and white PM separately [52,53]. Therefore, we aimed to address this gap in the literature by describing PM subcategories using national representative data collected by 24HDR.

Our results show considerable differences in intake across the four PM categories, ranging widest between sausages and bacon (Table 2). The low intakes of bacon were in the range of most countries in Europe [8]. Although mixed dishes were disaggregated into ingredients in the menuCH Survey, underestimation of minor ingredients such as bacon cannot be excluded [51].

Intakes of sausage were higher in the German-language region, while ham was preferred in the Italian-language region of Switzerland. These findings might reflect the different preferences and cultural influences of neighboring countries, such as high sausage intakes in Germany [14,17,54], where reported mean intake was 55 g/day and 26 g/day for men and women, respectively [13]. Intakes of other PM ranked second in quantity consumed by the population. This heterogeneous category contains foods (e.g., chicken nuggets) which are highly processed and increasingly criticized for their high energy and fat content [55]. 

In view of ongoing discussions concerning the safety of dietary nitrate/nitrite [56], P6M intake was also categorized based on presence or absence of nitrate/nitrite derived from curing (ham and bacon) or direct addition to meat mixtures (sausages) during production. Intakes of PM with nitrate/nitrite were threefold higher than intake without these substances. This estimate is conservative, since we excluded any PM intake where the presence of nitrate/nitrite was uncertain. These preservatives are used widely by the PM industry in Switzerland and Europe to stabilize color and improve microbiological safety as well as taste. Improvements in curing practices led to overall decreases in residual nitrite content of retail products in Europe and the United States [2,57]. To ensure adherence to legal limits of nitrate/nitrite, random controls of retail products are typically conducted [29]. In Switzerland, these results are available to the public but not used to develop a nitrate/nitrite database for PM and other foods, as in several countries of Europe and the USA [57]. This type of database in Switzerland would allow a more accurate estimate of PM contributions to total ingested nitrate/nitrite [57,58]. Based only on literature data, PM contribution of nitrates/nitrites is considered low in Switzerland compared with other sources and endogenous production [59]. Many substances have been considered, but no suitable replacement for nitrates/nitrites in PM has been found [60].

### 4.5. Processed Meat Consumption and Sociodemographic and Lifestyle Factors

The variable sex was shown to be a significant determinant of PM intake in the Swiss population. Indeed, the trend of higher PM consumption by men seems to be very widespread worldwide, as reported in Europe [4,8,14,52], the USA [15,61], and Australia [51]. Interestingly, PM intake was not associated with age in the current study, which agrees with UK population data [4] but differs from results of other studies in Europe [17,62]. Additionally, we did not observe an association between PM intake and income.

The positive association between PM intake and higher BMI is consistent with reports outside of Switzerland where waist circumference was also shown to determine PM intake [63,64]. BMI was recently reported as a significant predictor of overall meat consumption frequency in Switzerland [65]. This result might have important impacts on public health, since both overweight and obesity are recognized risk factors for many chronic diseases, also including cancers [49,63,66]. A stricter avoidance of high-fat and -energy foods during dieting might explain the negative association between PM intake and currently on a weight-loss diet. These results refer to the 5.4% of the Swiss population surveyed. The positive association between PM intake and current smoking was reported earlier elsewhere [62,64], but now confirmed for the Swiss population. The presence of two health risk factors (high BMI and smoking) might lead to an even higher risk because of potential synergistic effects [64]. Furthermore, an earlier report showed a higher prevalence of smoking in the Swiss population (28%), remaining stable for 10 years, compared with 23% in Table 1 [18]. This difference suggests healthy participation bias of the menuCH study group [67].

### 4.6. Processed Meat Consumption and Colorectal Cancer Risk 

The role of PM in cancer pathogenesis, in particular CRC, is still unclear but may involve the production of carcinogenic substances such as N–nitroso compounds (NOCs) formed from nitrite, or at high temperatures from heterocyclic aromatic amines (HAAs). Dietary factors such as protein and heme and high levels of fat may also intervene, leading to increased cancer risk, with evidence being strongest for the digestive tract [10,50,68,69,70]. Synergistic reactions of the abovementioned substances, interactions with whole diet, environmental exposures, individual differences in carcinogen-metabolizing enzymes, and gut microbiota may lead to CRC-promoting or preventive events [71,72,73,74].

According to the most recent Continuous Update Project report (CUP) by the World Cancer Research Fund, for every 50 g/day of PM intake, CRC risk increases by 16% [75]. Based on the current study, these risks might affect at least 33% and 13% of the menuCH study population who consumed at least 50 g/day or 100 g/day PM, respectively. Although 30% of the population did not report PM consumption at both 24HDR interviews, an additional questionnaire would be needed to conclude that these participants never consume PM. Examination of the ecological data for CRC incidence [76] revealed much higher cancer rates for men than women, over a period of 24 years, which is consistent with the observed higher PM intake in men than women. However, although there were regional differences in meat consumption (more UPM in the French-language region, more PM in the German-language region), CRC incidences did not differ between these regions. Further interpretations of PM intakes with regard to risk of CRC are currently not possible. More detailed and long-term data on PM intake are needed, and, ideally in the future, linkage of individual health data and food consumption in Switzerland.

### 4.7. Strengths and Limitations of the Study

This study provides a first in-depth description of PM intake and its subcategories, based on 24HDR and representative population data in Switzerland. The net response rate of 38% in the menuCH study is within the range of other national dietary surveys in Europe, ranging from 33% to 70% [77]. In the menuCH Survey, the main reasons for nonresponse were lack of time and interest [19]. The current results on total and PM types are crucial for public health policy and could also be used to estimate the percentage of the Swiss population who might be exposed to risks of cancer such as CRC [75,76], but further interpretations are not possible. The identified associations with unhealthy lifestyle factors could be used to improve public health measures. All consumption data were energy-standardized to account for overall differences in energy intake, and statistical weighting was applied on all studied variables to correct for the sampling design, nonresponse, and uneven distribution of dietary assessments over the week and season. This allowed a more accurate extrapolation of the results to the entire Swiss population.

Underestimation of PM intakes cannot be excluded due to dietary assessment errors, errors in PM categorization, and healthy participation bias [67]. Additional information at survey interviews such as PM brand name could increase the accuracy of categorizations and facilitate disaggregation of white and red PM intake [52]. Despite its importance in nutritional policy, the descriptive approach used in this study cannot capture aspects of whole diet with its complexity and interaction of nutrients and non-nutrients [20,54].

## 5. Conclusions

The present results provide a first quantitative basis for consumption of PM and its different subtypes. Considering the evidence-based health risks linked to PM consumption, a regular monitoring of PM intake should be done, and ideally separate from other meat types. This might be achieved by revising the questions on meat in the Swiss Health Survey [18]. Such data might impact nutritional policy, and lead to improvements in nutritional recommendations and public health interventions targeting unhealthy lifestyles. PM intake data for subgroups of the population will allow for better targeting nutritional recommendations toward specific sub groups. In addition to health concerns, increasing prioritization of sustainable nutrition will likely impact consumption levels of meat and PM in future [4,5]. Therefore additional questions will be needed in surveys to assess these and other emerging variables.

## Figures and Tables

**Figure 1 nutrients-11-02556-f001:**
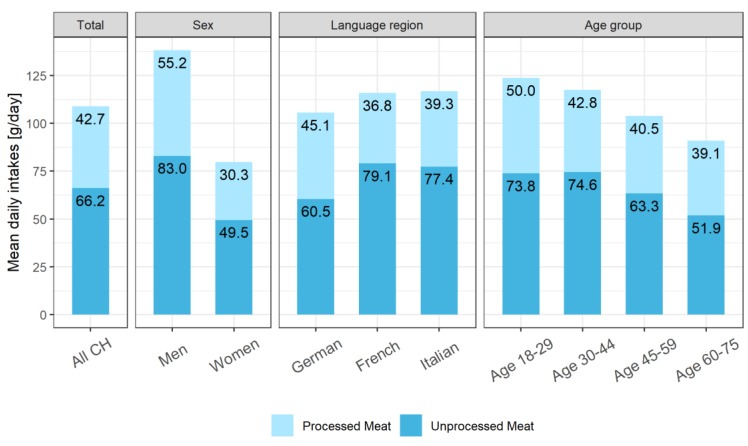
Comparison of mean daily intakes of processed and unprocessed meat for total population (All CH) and by sex, language region, and age group. All data weighted for sex, age, marital status, major area of Switzerland, household size, nationality, season, and weekday.

**Table 1 nutrients-11-02556-t001:** Description of total population and processed-meat consumers.

	Total Population		PM Consumers
	Crude	Weighted ^1^	Weighted ^1^
Number of participants	2057	4,627,878	3,331,232
Sex			
Men	45.4%	49.8%	54.9%
Women	54.6%	50.2%	45.1%
Language region ^2^			
German	65.2%	69.2%	71.0%
French	24.4%	25.2%	23.5%
Italian	10.4%	5.6%	5.5%
Age group ^3^			
18–29 years	19.4%	18.8%	18.0%
30–44 years	25.9 %	29.9%	31.0%
45–59 years	30.4%	29.8%	30.1%
60–75 years	24.3%	21.6%	20.9%
BMI category ^4^			
Underweight (BMI < 18.5 kg/m^2^)	2.4%	2.4%	2.0%
Normal (18.5 ≤ BMI < 25.0 kg/m^2^)	54.3%	54.1%	53.0%
Overweight (25.0 ≤ BMI < 30.0 kg/m^2^)	30.5%	30.6%	31.5%
Obese (BMI ≥ 30.0 kg/m^2^)	12.8%	12.9%	13.6%
Nationality			
Swiss	72.5%	61.4%	62.7%
Swiss binational	14.4%	13.8%	23.6%
Non-Swiss	13.0%	24.8%	13.7%
Education			
Primary	4.3%	4.7%	4.4%
Secondary	47.1%	42.6%	43.0%
Tertiary	48.5%	52.6%	52.4%
Household status			
Living alone	16.0%	18.1%	16.4%
Adult living with parents	7.7%	7.1%	7.3%
One-parent family with children	4.5%	4.4%	4.1%
Couple without children	33.5%	31.7%	31.4%
Couple with children	33.0%	32.8%	34.7%
Others	5.3%	5.7%	5.9%
Income CHF/month			
<6000	16.8%	17.7%	16.3%
6000–13,000	40.9%	39.8%	40.8%
>13,000	13.9%	14.9%	16.0%
Imputed ^5^	28.4%	27.6%	26.9%
Physical activity			
Low	14.7%	12.9%	17.0%
Moderate	22.1%	22.7%	21.2%
High	40.2%	40.3%	40.6%
Imputed ^5^	23.0%	24.2%	21.2%
Smoking status			
Never	44.5%	42.9%	40.8%
Former	33.5%	33.6%	34.4%
Current	22.0%	23.3%	24.5%
Health status			
Very bad to medium	13.2%	12.7%	13.0%
Good to very good	86.6%	87.1%	86.7%
Currently on a diet			
No	94.3%	94.4%	95.1%
Yes	5.5%	5.4%	4.6%

^1^ Weighted for sex, age, marital status, major area of Switzerland, household size, nationality. ^2^ German-language region: cantons Aargau, Basel–Land, Basel–Stadt, Bern, Lucerne, St. Gallen, Zurich; French-language region: Geneva, Jura, Neuchâtel, Vaud; Italian-language region: Ticino. ^3^ Age corresponds to self-reported age on the day of the 24 h dietary recall interview. ^4^ BMI based on measured height and weight; self-reported estimations were used when not possible to measure (e.g., lactating and pregnant women). ^5^ Multivariate imputation by chained equations was used for missing values; imputed values of less than 0.4% are not shown. PM: processed mean; CHF: Swiss francs; BMI: body mass index.

**Table 2 nutrients-11-02556-t002:** Mean daily intake for total and main categories of processed meats for the population (all CH) and by language region, sex, and age group (g/day; g/1000 kcal).

	Ham			Bacon			Sausages			Other Processed Meat			Total Processed Meat		
	Mean (g/day)	SEM	Mean (g/1000 kcal)	Mean (g/day)	SEM	Mean (g/1000 kcal)	Mean (g/day)	SEM	Mean (g/1000 kcal)	Mean (g/day)	SEM	Mean (g/1000 kcal)	Mean (g/day)	SEM	Mean (g/1000 kcal)
**All CH**	10.3	0.4	4.6	2.0	0.2	0.8	18.1	0.7	7.8	12.3	0.7	5.4	42.7	1.2	18.7
**Language region**															
German	10.4	0.6	4.5	2.0	0.2	0.8	19.4	0.9	8.3	13.3	0.9	5.7	45.1	1.5	19.4
French	8.7	0.7	4.2	2.1	0.3	0.9	16.4	1.6	7.1	9.6	1.2	4.3	36.8	2.2	16.4
Italian	16.2	1.7	8.8	1.7	0.4	0.8	10.3	1.6	4.7	11.2	2.0	5.8	39.3	3.2	20.1
**Sex**															
Men	13.0	0.7	5.3	2.5	0.3	0.9	23.4	1.2	9.3	16.2	1.3	6.5	55.2	2.0	21.9
Women	7.5	0.5	4.0	1.6	0.2	0.8	12.9	0.8	6.4	8.3	0.7	4.3	30.3	1.2	15.4
**Age group**															
18–29 years	11.4	1.2	4.5	1.6	0.3	0.6	18.4	1.8	7.3	18.6	2.0	8.0	50.0	3.0	20.4
30–44 years	9.6	0.8	4.3	2.2	0.3	0.8	20.1	1.6	8.4	10.9	1.3	4.7	42.8	2.3	18.3
45–59 years	10.8	0.8	5.1	2.1	0.3	0.9	17.6	1.3	7.6	9.9	1.1	4.2	40.5	2.0	17.7
60–75 years	9.3	0.9	4.6	2.0	0.4	1.0	16.0	1.3	7.7	11.8	1.3	5.6	39.1	2.3	18.9

All data are weighted for sex, age, marital status, major area of Switzerland, household size, nationality, season, and weekday; SEM: standard error of the mean.

**Table 3 nutrients-11-02556-t003:** Mean daily intake of processed meats with or without the preservatives nitrate/nitrite for the total population (all CH) and by language region (g/day).

	Nitrate/Nitrite		Without Nitrate/Nitrite	
	Mean (g/day)	SEM	Mean (g/day)	SEM
All CH	27.4	0.9	9.8	0.6
Language region				
German	28.9	1.1	10.6	0.9
French	23.4	1.7	7.5	1.0
Italian	26.6	2.5	10.6	1.8

All data are weighted for sex, age, marital status, major area of Switzerland, household size, nationality, season, and weekday. Undefined products were excluded (5.5 g/day): undefined sausages, Adrio (CH), Terrines, Netzbraten, meat balls, Selzacher Umgangspastete (CH); SEM: standard error of the mean.

**Table 4 nutrients-11-02556-t004:** Associations between processed meat intake and sociodemographic and lifestyle factors, by multivariable linear regression analyses (*N* = 2057).

	Processed Meat (g/1000 kcal)	
	Adjusted Coefficients ^1^	95% CI
Sex		
Men	0	ref.
Women	**−4.7**	**[−6.7; −2.7]**
Language region ^2^		
German	0	ref.
French	**−2.9**	**[−5.2; −0.7]**
Italian	0.2	[−4.0; 4.4]
Age group ^3^		
18–29 years	1.1	[−2.2; 4.4]
30–44 years	0	ref.
45–59 years	−1.8	[−4.3; 0.7]
60–75 years	−1.3	[−4.4; 1.8]
BMI category ^4^		
Underweight (BMI < 18.5 kg/m^2^)	1.2	[−5.2; 7.6]
Normal (18.5 ≤ BMI < 25 kg/m^2^)	0	ref.
Overweight (25 ≤ BMI < 30 kg/m^2^)	**3.9**	**[1.6; 6.1]**
Obese (BMI ≥ 30 kg/m^2^)	**5.5**	**[2.4; 8.7]**
Nationality		
Swiss	0	ref.
Swiss binational	1.1	[−1.7; 3.9]
Non-Swiss	−1.0	[−3.4; 1.4]
Education degree		
Primary	1.9	[−2.8; 6.6]
Secondary	0	ref.
Tertiary	**−2.9**	**[−5.0; −0.8]**
Household status		
Living alone	−0.4	[−3.5; 2.7]
Adult living with parents	3.4	[−1.2; 7.9]
One-parent family with children	−2.0	[−7.0; 3.0]
Couple without children	0	ref.
Couple with children	0.5	[−2.0; 3.1]
Others	1.7	[−2.9; 6.2]
Income (CHF/month)		
<6000	0.5	[−2.7; 3.7]
6000–13,000	0	ref.
>13,000	0.0	[−2.9; 2.9]
Physical activity level		
Low	0	ref.
Moderate	−2.3	[−5.5; 1.0]
High	−2.5	[−5.4; 0.3]
Smoking status		
Never	0	ref.
Former	1.8	[−0.4; 4.0]
Current	**3.1**	**[0.6; 5.6]**
Health status		
Very bad to medium	0.82	[−2.2; 3.9]
Good to very good	0	ref.
Currently on a diet		
Yes	**−7.5**	**[−11.7; −3.4]**
No	0	ref.

^1^ Adjusted for the variables shown above; weighted for sex, age, marital status, major area, household size, nationality, season, and weekday. ^2^ German-language region included cantons: Aargau, Basel–Land, Basel–Stadt, Ben, Lucerne, St. Gallen, Zurich; French-language region: Geneva, Jura, Neufchâtel, Vaud; and Italian-language region: Ticino. ^3^ Age corresponds to self-reported age on the day of the 24 h dietary recall interview. ^4^ BMI (body mass index) was based on measured height and weight; self-reported estimations were used when not possible to measure (lactating and pregnant women or people with disabilities). Coefficients in bold are associated with a *p*-value < 0.05. CHF: Swiss francs; CI: Confidence interval.

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
