# Peer review of "Intake of Processed Meat and Association with Sociodemographic and Lifestyle Factors in a Representative Sample of the Swiss Population"

_nutrients, 2019, doi:10.3390/nu11112556_

Round 1

Reviewer 1 Report

This paper provides some fruitful information for processed meat consumption in Swiss population, and I believe this paper could interest some readers.

I have a few comments:

1, This paper focuses on processed meat consumption. But what happens to unprocessed meat? Could you give some figures on it. I believe you have the information.

2,The policy implications are not detailed. It is better to give some policy implications.

3, Table 3, the term "Nitrite/Nitrate" is confusing. Is it implies "meat with Nitrite/Nitrate" or "intake of Nitrite/Nitrate"  

Reviewer 2 Report

Manuscript ID:  Nutrients:  Intake of Processed Meat and Association with Sociodemographic and Lifestyle Factors in a Representative Sample of the Swiss Population

Summary: This paper describes a cross-sectional study using nationally representative data in Switzerland to quantify intake of processed meats, and evaluate associations between sociodemographic and lifestyle factors with processed meat intake among 2,057 Swiss adults.   Although participation in the survey was quite low (38%), the authors found that processed meat was consumed by nearly 72% of the Swiss population, with lower intakes among women (vs. men) and in French-speaking (vs. German-speaking) regions. Processed meat intake was also higher among the overweight and obese compared to normal BMI counterparts, and among current smokers compared to never smokers.   This descriptive paper is generally well-written and provides some evidence of differences in processed meat intake across Switzerland and associations between processed meat intake with unhealthy lifestyle factors (obesity/smoking).  However, the authors’ conclusion re: emphasizing the importance of processed meat falls short given that there are other unhealthy factors linked to processed meat intake and health outcomes that were not comprehensively considered in the study.  For example, it would be helpful to understand correlations between processed meat intake and other dietary indicators, which may also be important for health and linked to unhealthy lifestyle factors.

 I have provided my detailed comments for authors to consider below.

Major Comments:

Introduction.  I’d prefer to see more mention of the current evidence from other studies of sociodemographic/lifestyle factors and processed meat intake in other populations that could be cited here.  If there is a lack of studies examining associations, perhaps there are studies of unprocessed meat and determinants that could be discussed here.  In making the argument for the prioritization of understanding determinants of processed meat intake in Switzerland, it would be helpful to include more information on what is known from other studies in other populations.  If there is not clear evidence from other studies, authors should state this. Also, the authors should provide some justification for why the selected sociodemographic/lifestyle factor were considered.

Abstract, line 30.  It is unclear how associations of processed meat with BMI and smoking warrant the “need for continued monitoring of PM intake, separate from other meat types” given that associations were not adjusted for unprocessed meat intake and unprocessed meat intake was higher than processed meat intake in this population.  Thus, it is difficult from these analyses to determine the importance of PM specifically.

Minor Comments:

Abstract, first sentence.  Would be helpful to give examples of specific health risks linked to processed meat here.  Also, how much of a burden are these health risks in Switzerland?

Abstract, results lines 22-25.  These results are difficult to interpret unless the reader is already familiar with average intake of processed meats in other parts of the world.  I’d recommend starting with a sentence that describes overall processed meat intake patterns so readers can assess how findings in Switzerland compare to other developed/developing countries.

Abstract, results lines 27-29.  Specify that this is a difference measure in description of findings by BMI and smoking. 

Introduction, line 56.  Should this be “determinants” instead of “determination”?

Materials and Methods, Study Design, line 76.  Low participation (38%) should be addressed. Is there any national data available to compare those included in the study vs. non-participants?
